# Prevalence and intensity of soil-transmitted helminths infection among individuals in model and non-model households, South West Ethiopia: A comparative cross-sectional community based study

**Yonas Alemu** [1]*, **Teshome Degefa** [2], **Mitiku Bajiro** [2], **Getachew Teshome** [2]

1 Department of Microbiology, Immunology, and Parasitology (DMIP), School of Medicine, Addis Ababa University, Addis Ababa, Ethiopia, 2 School of Medical Laboratory Sciences, Institute of Health, Jimma University, Jimma, Ethiopia

* alemuyonas@yahoo.com, yonas.alemu@aau.edu.et

## Abstract

Soil-transmitted helminths (STH) is a term used to refer to infections caused by intestinal worms mainly due to *A. lumbricoides*, *T. trichiura*, and hookworm species which are transmitted through contaminated soil. This study was conducted to assess the prevalence and intensity of STHs infection among individual members living within the selected household heads (HHs) certified either as a model HHs or non-model HHs based on the implementation level of a training program known as the Health Extension Program (HEP). A community-based comparative cross-sectional study was conducted from April to June 2018 at Seka Chekorsa Woreda, Jimma zone. Model and non-model HHs were selected systematically from each of the randomly selected district villages employing a multistage sampling technique. Sociodemographic and risk factors data associated with STHs infections were collected using a pre-tested structured questionnaire. Parasitological stool sample microscopic examination was done using saline wet mount and Kato Katz thick smear technique. Data analysis was performed using SPSS software version 20 for descriptive statistics, comparison, and logistic regression at a p-value < 0.05 for statistical significance. Overall, 612 individuals were recruited in the study from 120 randomly selected HHs. The prevalence of STHs infections was found to be 32.4%. A total of 45 (14.7%) model and 153 (50.0%) non-model individual participants were positive for at least one species of STHs showing a significant difference between individuals in model and non-model HHs (AOR: 6.543, 95% CI; 4.36–9.82, P<0.001). The dominant STHs were *T. trichiura* (21.6%) followed by *A. lumbricoides* (6.4%) and hookworms (2.3%). The intensity of *T. trichiura and A. lumbricoides* infection have shown a significant difference (p<0.05) while hookworm species infection was not significantly different (p>0.05) for the individuals in the HHs groups. On the other hand, the households training status, age of participants, and latrine use pattern were found significant predictors of STHs infection prevalence in the multivariate analysis

**Data Availability Statement:** All relevant data are within the paper.

**Funding:** The author(s) received no specific funding for this work.

**Competing interests:** The authors have declared that no competing interests exist.

(P<0.05). Therefore, the prevalence and intensity of STHs infection was higher among individuals living in a non-model HHs than model HHs.

## Introduction

Soil-transmitted helminths (STHs) infection due to *A. lumbricoides*, *T. trichiura*, and hookworm species are the most common parasitic infections worldwide but occur in the greatest numbers in Sub-Saharan Africa, East Asia, China, India, and South America [1,2]. About 2 billion people are infected with one or more STHs species. More than 4 billion people are at risk of infection, 135,000 people die a year and 4.94 million years lived with disability attributable due to STHs. Globally, about 819, 439, and 465 million people were infected with *A. lumbricoides*, hookworm species, and *T. trichiura*, respectively. Over 450 million, mostly children suffer from significant morbidity; 44 million pregnant women suffer clinical effects from hookworm-associated anemia and also a severe impact on the elders [3,4].

STHs are found most prevalently throughout the tropics and subtropics wherever hygiene is poor, safe water and sanitation facilities are lacking and health services are insufficient. They are transmitted via ingestion or skin penetration of the infective stages [5,6]. Their effects on health include anemia delays in physical growth and cognition, decreased stamina and work output, and complications during pregnancy [7]. Morbidity is related to the number of worms harbored. People with light infections usually have no symptoms while heavier infections can cause a range of symptoms including diarrhea, abdominal pain, malaise and weakness, intestinal blood losses, loss of appetite, reduction of nutritional intake, and physical fitness [8,9].

STHs infections are often overdispersed in endemic communities, such that most individuals harbor just a few worms in their intestines while a few hosts harbor disproportionately large worm burdens. There is also evidence of familial and household aggregation of infection [3,10]. Control of STHs can be achieved by the targeted use of chemotherapy and improvement of sanitation, drinking water, use of pit latrines instead of open defecation, and good hygiene practices. World Health Organisation (WHO) recommends mass drug administration with Albendazole 400mg and Mebendazole 500mg to all people at risk of infection living in endemic areas disregard of diagnosis [11,12].

Additionally, an effective, targeted and simple type of health education is recommended as a first option to create an enabling environment for other strategies to thrive, especially in underprivileged communities. Thus, participation of the community represents one of the cardinal tools of disease control programs as improvements in awareness and understanding can greatly increase the realization and sustainability of long-term STHs control strategies. However, the success of control initiatives involving the community depends on the level of the communities' uptake of the program, which is linked to the understanding of the community's knowledge, practices, and perceptions of the disease found to be instrumental in designing and implementing effective community-based programs [13–15].

As a community involving health intervention, the Health Extension Program (HEP) launched by the Ethiopian Federal Ministry of Health in 2003 includes 16 essential health packages categorized under four major program areas. These are hygiene and environmental sanitation; disease prevention and control; family health services; and health education and communication. Those HHs that are trained and successfully implement at least 75% of these training packages become certified model HHs and their health status is also assumed to be superior to non-model HHs [16–18]. Taking the training packages concept into account, there

is insufficient empirical evidence to support whether the implementation of the HEP in Ethiopia brought an impact on the prevalence of sanitation and hygiene-related infections such as STHs. Therefore, the objective of this study is to assess the prevalence & intensity of STHs infection among individuals in HEP model & non-model HHs.

## Methods and materials

### Study area

This study was conducted in Seka Chekorsa woreda, which is found in the Jimma zone of Oromia regional state, southwest Ethiopia. It is located 366 km away from the capital city Addis Ababa and 20 km away from Jimma town. It is bounded by Gomma and Manna woreda in the north, Gera woreda in the south, Dedo woreda and Jimma Town in the East, and Shabe Sombo woreda in the west. This woreda covers an estimated area of 455km$^2$ and has 36 districts (34 rural and 02 urban; of which 29 districts are models while 07 are non-models), has 01 Hospital, 09 Health centers, 35 Health posts, and 84 health extension workers.

The altitude of this woreda ranges from 1580 to 2560 meters above sea level and rainfall ranges from 1,200 to 2,800 mm. The minimum and maximum daily temperatures of the area are 12.6˚C and 29.1˚C, respectively. Perennial rivers include the Abono, Anja, Gulufa, and Meti. A survey of the land in this woreda shows that 45.3% is cultivable (44.9% was under annual crops), 6.1% pasture, 25.8% forest, and the remaining 22.8% are considered swampy, degraded, or otherwise unusable. Khat, peppers, fruits & teff are important cash crops including Coffee plantations with over 50 km$^2$ area [19]. The 2007 national census reported the total population for this woreda is 208,096, of whom 104,758 were men and 103,338 were women; 7,029 (3.38%) of its population were urban dwellers and around 90% of the residents are farmers.

### Study design and period

A community-based comparative cross-sectional study was conducted from April to June 2018.

### Source and study population

All HHs selected from the districts in Seka Chekorsa woreda were the source population. Whereas, all individual members in HHs permanently living in the study area at least for 6 months, aged greater than two years, who were voluntary to provide consent to participate in the survey, could provide stool samples and did not take anti-helminthic treatment for 28 days before data collection were included in the study.

### Sample size and sampling technique

The total sample size for this study was calculated by Epi-Info Version 7 statistical software (Stat calc) using the double population proportion formula to detect the difference between individuals among model and non-model HHs of 26.05% and 52.1% [20], respectively. With a confidence level of 95%, considered a power of 80%, 1:1 ratio, OR of 2 or with an assumption of 50% reduction, non-response of 10%, and design effect of 2. Therefore, the final sample size was 612 (306 model and 306 non-model individual members living within the HHs).

A multistage sampling technique was applied to select individuals living in the HHs from the villages found in the districts of Seka Chekorsa woreda. In the first stage, from the total of 36 districts in the woreda that were categorized as either model (n = 29) or non-model (n = 7), 04 districts were randomly selected (Buyo Kechema and Kusaro from model districts and

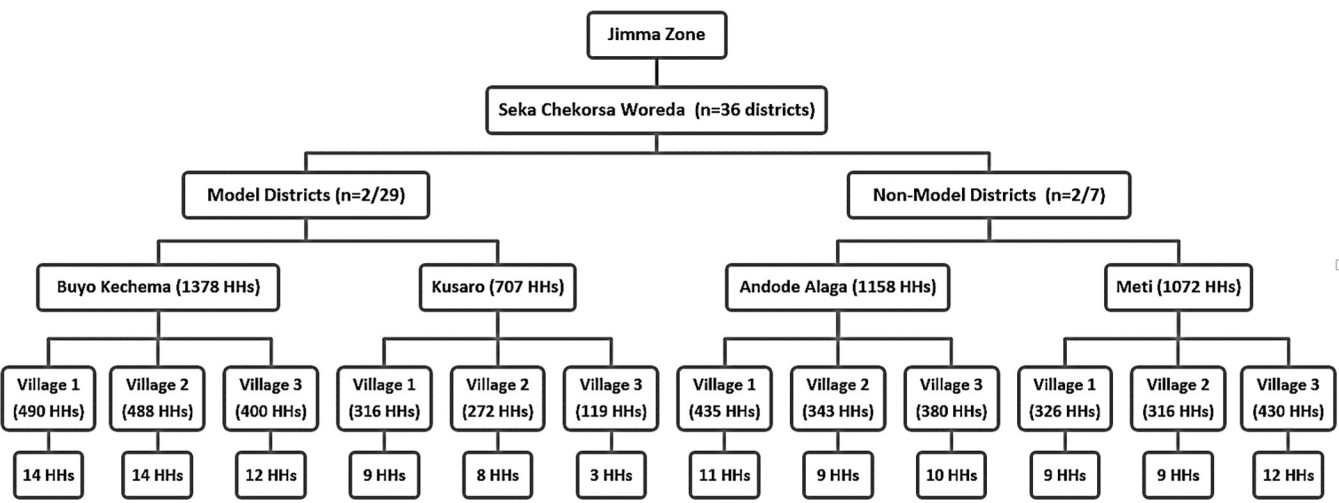

**Fig 1. Diagrammatic representation of the multi-stage sampling technique used to select individuals in model and non-model HHs. Key:** HHs-Household heads.

Andode Alaga and Meti from the non-model districts). In the second stage, all villages (n = 3) (locally referred to as "zones") within each of the selected districts were selected. Finally, considering the mean family size of the Jimma zone which is 5.1±1.8, and a total sample size of 612 individual members, 120 HHs were enrolled with systematic sampling. Calculated HHs number was allocated proportionally according to the HHs size within each village (Fig 1).

## Study variables

The STHs infection and intensity were the dependent variables while the household training status (HH) (model or non-model), Socio-demographic, Socio-economic characteristics, Personal hygiene, environmental sanitation, waste disposal, and awareness about STHS infections related questions were our independent study variables.

## Data collection

Sociodemographic characteristics and risk factors that predispose to STH infection were collected using pre-tested structured questionnaires. Trained data collectors administered the questionnaires that are prepared in English and translated into Afan Oromo language.

## Stool sample collection and processing

All individuals were provided labeled stool cups and instructed to bring sufficient stool samples. All specimens were checked for their label and quantity. Direct saline wet mount stool examination technique was performed by emulsifying a small portion of stool sample with normal saline for the microscopic examination using light microscopy at the nearby health post/health center located in each district immediately. Then, aliquots of each specimen were transported to the medical parasitology teaching laboratory of the school of medical laboratory sciences, Jimma University using the cold box for the same-day preparation of the Kato-Katz thick smear technique.

Screening of STHs eggs was based on a 41.7 mg Kato Katz template to determine the parasite's egg per gram in the stool (EPG) by calculating the number of eggs counted multiplied by 24 [21]. Infection intensity was then categorized as light, moderate, and heavy infection for

common STHs infections following the WHO standard procedure [22]. Experienced laboratory technologists performed the laboratory procedures according to the standard operating procedures. Stool samples were randomly selected for quality control and examined by a third person who was blinded to the previous test results.

## Data analysis and interpretation

Questionnaires were checked for completeness, data were entered into MS excel, cleaned, and imported to SPSS version 20 for statistical analysis. Descriptive analysis including frequency, mean, and percentage was used to summarize the demographic characteristics of the study participants. The Chi-square test was calculated to observe associations of variables. The geometric mean intensity of the parasite EPG of stool was calculated for all infected and non-infected individuals by incorporating zero counts during the analysis adjusted by adding one to each datasets and removing from the algorithmic results. Independent samples t-test was used to compare the mean difference in intensity of STHs infection between the two households. Bivariable analysis was computed to see the association of each independent variable with the dependent variable. Candidate variables for multivariable analysis were selected when the P-value was less than 0.25 in bivariate analysis. In all comparisons, a P-value of <0.05 was considered statistically significant.

## Ethical statement

Ethical clearance and letter of permission were obtained from the Institutional Review Board of Jimma University (IRB letter number, IHRPG/281/2018) and Official permission was sought from Health offices. Written informed consent to participate in the study was obtained from all the adults and assent was sought from all participating children before conducting an interview or collecting a stool sample. Confidentiality of their information was maintained and participants who become positive for any intestinal parasites were linked to the health institutions to be treated according to the Ethiopian drug administration guideline [23].

## Results

### Socio-demographic, economic characteristics, and hygiene conditions

Overall, a total of 612 individuals were included in the study from the total of 120 HHs visited during the study period. About half, 308 (50.33%) of the study participants were males and most of the study participants 199 (32.52%) were in the age group of 5–14 years, followed by 15–24 years 104 (16.99%). The mean age of study participants was 22.35 ±16.72 years in the range of 2–70 years. There were no significant differences between individual participants in model and non-model HHs in terms of gender ($X^2 = 0.654$, p = 0.419) and age ($X^2 = 0.426$, p = 0.514). The mean age for HHs was 40.83±12.041 with the mean family size of 5.16±1.914. The majority of HHs were able to read and write 84 (70%) and farmers 81 (67.5%). Most of the HHs, 54 (45%) of them earn an annual income of 1000–3000 birr and live in earthen mud-plastered houses 112 (93.3%). Annual family income showed significant differences between model and non-model HHs (p < 0.05) (Table 1).

Latrine availability to the study participants accounts for (98.3%) of out which (63.3%) of the latrines had a lid cover and about (68.3%) with handwashing facilities near the latrines. The majority of them use spring water (90.8%) as a source of drinking water and then piped water (9.2%). The use of pit waste disposal accounts (53.3%) while in the open field was (46.7%). Whereas, an individual's shoe-wearing status and nail hygiene cleanness account for (92.65%) and (95.59%), respectively. A total of 78 (65%) HHs heard and knew about STHs

**Table 1. Socio-demographic, socio-economic and hygiene conditions among individuals in model & non-model HHs, Seka Chekorsa woreda, Jimma, Southwest Ethiopia.**

| Variables | Model (n = 306) No (%) | Non-model (n = 306) No (%) | $X^2$ | p-value |
|---|---|---|---|---|
| **Gender** | | | | |
| Male | 149 (48.7) | 159 (52.0) | 0.654 | 0.419 |
| Female | 157 (51.3) | 147 (48.0) | | |
| Age | | | | |
| <15 | 137 (44.8) | 129 (42.2) | 0.426 | 0.514 |
| ≥15 | 169 (55.2) | 177 (57.8) | | |
| **Family size (n = 120)** | | | | |
| <5 | 24 (40.0) | 24 (40.0) | 0.00 | 1.00 |
| ≥5 | 36 (60.0) | 36 (60.0) | | |
| **Educational status (n = 120)** | | | | |
| Unable to read and write | 16 (26.7) | 20 (33.3) | 0.635 | 0.426 |
| Able to read and write | 44 (73.3) | 40 (66.7) | | |
| **Occupation (n = 120)** | | | | |
| Farmer | 44 (73.3) | 37 (61.7) | 5.236 | 0.155 |
| Housewife | 15 (25.0) | 19 (31.7) | | |
| Merchant | 0 (0.0) | 3 (5.0) | | |
| Daily laborer | 1 (1.7) | 1 (1.7) | | |
| **Family income (n = 120)** | | | | |
| <1000 | 8 (13.3) | 13 (21.7) | 15.731 | 0.001* |
| 1000–3000 | 19 (31.7) | 35 (58.3) | | |
| >3000 | 33 (55.0) | 12 (20.0) | | |
| **Mud-plastered (n = 120)** | | | | |
| Mud-plastered, earthen | 54 (90.0) | 58 (96.7) | 2.236 | 0.135 |
| Stone walls with cement | 6 (10.0) | 2 (3.3) | | |
| **Latrine availability (n = 120)** | | | | |
| Yes | 59 (98.3) | 59 (98.3) | 0.000 | 1.000 |
| No | 1 (1.7) | 1 (1.7) | | |
| **Latrine lid covered (n = 120)** | | | | |
| Yes | 50 (83.3) | 26 (43.3) | 20.670 | <0.001* |
| No | 10 (16.7) | 34 (56.7) | | |
| **Hand washing facility (n = 120)** | | | | |
| Yes | 51 (85.0) | 31 (51.7) | 15.404 | <0.001* |
| No | 9 (15.0) | 29 (48.3) | | |
| **Water source (n = 120)** | | | | |
| Pipe | 11 (18.3) | 0(0.0) | 12.110 | 0.001* |
| Spring | 49 (81.7) | 60 (100.0) | | |
| **Waste disposal (n = 120)** | | | | |
| Pit | 42 (70.0) | 22 (36.7) | 13.393 | <0.001* |
| Open field | 18 (30.0) | 38 (63.3) | | |
| **Shoe wearing** | | | | |
| Yes | 279 (91.2) | 288 (94.1) | 1.943 | 0.163 |
| No | 27 (8.8) | 18 (5.9) | | |
| **Nail hygiene cleanness** | | | | |
| Yes | 297 (97.1) | 288 (94.1) | 3.138 | 0.076 |
| No | 9 (2.9) | 18 (5.9) | | |

Key

*- statistically significant at p-value < 0.05, n- sample size, No.- Number, $X^2$-chi-square.

locally named 'Raammoo garaa' explaining the most associated symptoms are abdominal pain (30%) and diarrhea (28.3%). The contaminated hand was perceived as the most common mode of transmission (49.17%) and deworming was a preventive measure among 41.67% of HHs (Table 1).

Availability of latrine lid cover, hand washing facilities near the latrine, source of water, and waste disposal system has shown a significant difference (P<0.05) between model and non-model HHs. Similarly, HHs' awareness on source of STHs infection, mode of transmission, and prevention including signs and symptoms showed a statistically significant difference among the two groups of HHs (P<0.05) (Table 1).

### Prevalence of soil-transmitted helminths

The prevalence of STHs infection among the study participants who provided a stool sample was 198 (32.35%), out of which 45(14.71%) were among model and 153(50.0%) in non-model HHs, respectively. Whereas, the overall prevalence of intestinal parasitic infections was 212 (34.64%), 50(16.34%) among model and 162(52.94%) non-model, respectively. There was a statistically significant difference in the prevalence of STHs (OR = 5.8, 95% CI; 3.936–8.547, P<0.001) and overall intestinal parasites (OR = 5.76, 95% CI; 3.95–8.399, P<0.001) between model and non-model individual members in the HHs.

STHs parasites identified among the study participants were namely *T. trichiura 132* (21.57%), *A. lumbricoides 39* (6.37%), and hookworm species 14 (2.29%) in a single infection. Whereas mixed infections were found as double STHs in 8 (1.31%), triple STHs in 1 (0.16%), and STHs mixed with other intestinal parasite species in 5 (0.82%) (Fig 2). The difference in the prevalence of *A. lumbricoides* and *T. trichiura* was found statistically significant between model & non-model participants (P<0.001). Regarding the mixed infection status, 43(14.1%), 1(0.33%), and 1(0.33%) model individual members in the HHs had single, double, and triple STHs infections while 146(47.71%) and 7(2.29%) non-model individual members in the HHs had a single and double STHs infections, respectively. The status of mixed STHs infection was also statistically significant with the status of HHs (P<0.001). The other intestinal parasites

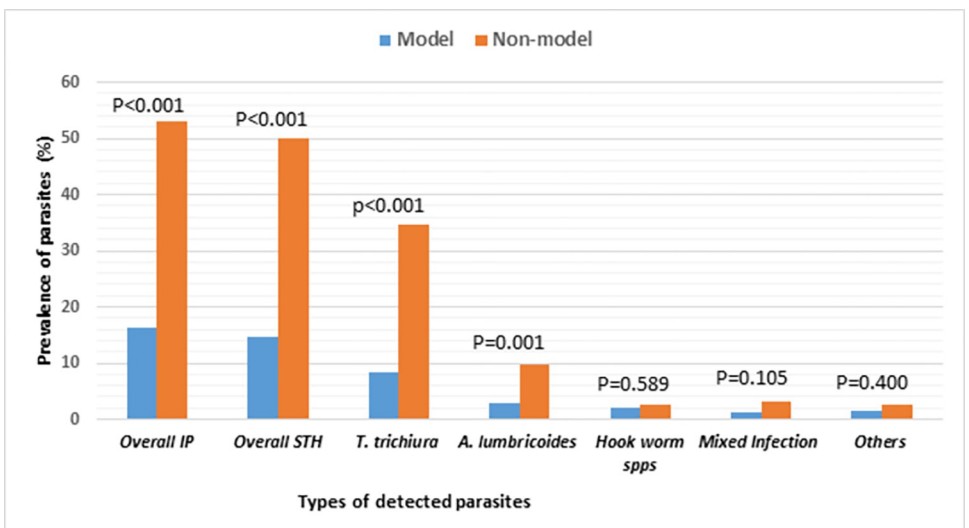

**Fig 2. Prevalence of STHs and mixed parasitic infections among individual participants of model and non-model HHs, Seka Chekorsa woreda, Southwest Ethiopia. Key:** IP: Intestinal parasites, others: (*E. vermicularis*, *E. histolytica* /*dispar*, and *Taenia* species), mixed infection: Multiple infections of STHs parasites or with any of other types of parasites.

detected in stool samples were *E. vermicularis* 7 (1.14%), *Taenia species* 2 (0.33%), and tropho-zoite of *E. histolytica / E. dispar* 4 (0.65%).

The prevalence of STHs infections between male 96(31.17%) and female study participants 102 (33.55%) did not show a significant difference (P>0.05). The prevalence of STHs parasites among the age group >15 years was found at 101(51.01%) which has no significant difference (P>0.05) from the age group of <15 years 97(49.09%). Of the 322(52.61%) individual members in the HHs trained by HEWs on HSEP, 58(18.01%) were infected and from untrained 290 (47.39%) individual members in the HHs, 140(48.28%) were infected with STHs showing a significant difference of the STHs infection in training status (P<0.001).

## The intensity of soil-transmitted helminths

The intensity of STHs infection was categorized based on the WHO classification thresholds using Kato-Katz thick smear method of parasites egg quantification expressed in eggs per gram (EPG) of stool. The geometric mean intensity of *T. trichiura* infection in our study was 2.30 (EPG ranging from 0–1,368), *A. lumbricoides* with 0.61 (EPG ranging from 0–31,840), and hookworm species with 0.13 (EPG ranging from 0–420). There was no statistically significant difference (P>0.05) in the mean intensity of hookworm species infection between model and non-model individual members in the HHs although *A. lumbricoides* and *T. trichiura* infection were having significant difference for the two groups (p<0.05) (Table 2).

The mean intensity of *T.trichiura* and *A.lumbricoides* was high in the age group of ≤15 years and ≥15 years for hookworm species. Whereas, the mean intensity of *A. lumbricoides* was higher among male study participants while hookworm species and *T. trichiura* were high among female participants. However, the mean intensity of STHs infection has no significant difference with the age and gender of the study participants (P>0.05).

## Risk factors of soil-transmitted helminths

Logistic regression analysis was performed to observe whether the overall STHs infection was significantly associated with the potential risk factors. Independent variables of P≤0.25 in Binary logistic regression analysis were selected as potential candidates for multiple logistic

**Table 2. Infection intensity thresholds of STHs with the HH training status of selected districts of Seka Chekorsa woreda, Jimma zone, Southwest Ethiopia.**

| STHs species | STHs Infection status | | HH status | | P-value |
|---|---|---|---|---|---|
| | | | Model | Non-model | |
| *Trichuris trichiura* | Geometric mean (EPG) | | 0.56 | 5.99 | <0.001* |
| | Intensity class | Light | 28(19.9) | 108(76.6) | |
| | | Moderate | 0(0.0) | 05(3.6) | |
| | | Heavy | 0(0.0) | 0(0.0) | |
| *Ascaris lumbricoides* | Geometric mean (EPG) | | 0.26 | 1.06 | <0.001* |
| | Intensity class | Light | 12(23.5) | 35(68.6) | |
| | | Moderate | 01(2.0) | 03(5.9) | |
| | | Heavy | 0(0.0) | 0(0.0) | |
| **Hookworm species** | Geometric mean (EPG) | | 0.10 | 0.16 | 0.341 |
| | Intensity class | Light | 07(43.8) | 09(56.2) | |
| | | Moderate | 0(0.0) | 0(0.0) | |
| | | Heavy | 0(0.0) | 0(0.0) | |

Key

*- statistically significant at p-value < 0.05.

regression analysis using the backward stepwise method with Hosmer-Lemeshow goodness-of-fit statistics. A p-value <0.05 was considered a risk factor associated with the STHs infection. After adjusting for confounding variables, the present study reported that the distribution of STHs infections varies between individual members in the model and non-model HHs. The finding showed that individuals in the non-model HHs (AOR: 6.543, 95%CI; 4.36–9.82, p<0.001) were significantly 6.5 times more likely infected with STHs than those in model HHs (Table 3).

Regarding STHs infection with the age of individual members in the HHs, indicates that those individuals aged below 15 years (AOR: 1.515, 95%CI; 1.04–2.21, p = 0.030) were showing a significant finding of 1.5 times more likely infected with STHs infection than individuals aged above 15 years. The other significant finding was the latrine use pattern of individual members in the HHs, those individuals using a latrine sometimes were 2.8 times more likely to be infected with the STHs infection compared to the individuals using a latrine always. Whereas, those individual members of HHs with no shoe wearing and handwashing habits, dirt on a fingernail, not owning a latrine, not having a handwashing facility near a latrine and without a lid, open field defecating, and disposing of waste in the open field were more likely infected with STHs than individual members in the HHs with the constant variables. However, these independent variables were not significantly associated (p>0.05) with the STHs infection in multivariate analysis (Table 3).

## Discussion

This study is the first to provide information comparing the difference in the prevalence and intensity of STHs infection among individual members in the model and non-model HHs in Ethiopia. Regardless of the intensive efforts to train HHs with the HEP agendas having possible indirect impact on hygiene related infections, the STH's overall prevalence in the study area before and after the implementation of the health extension program was unknown. The current study indicates the prevalence of STHs infection among individuals living in model and non-model families in this rural community to be 32.35%. However, there was a significant difference in the prevalence with the HH status with a reduction of prevalence for those trained and certified on health extension packages. The reduction in prevalence may thus related to the behavioral change to seek healthcare service, the improvement in the hygiene and sanitation conditions after the training or else the existing economic status difference of the households that might reduce their exposure to STHs infection.

The finding of STHs prevalence from our study was found lower when compared to studies conducted in Ecuadorian Amazon, Nigeria, and Ethiopia with the prevalence reports of 52.1% [20], 65% [24], 46.9% [25], 84.24% [26] and 67.3% [27]. Whereas, our STHs infection prevalence was slightly higher than a study in Jimma 20.9% [28] and comparable with 32.5% [29] in Cameroon. These differences may be due to the effect of HEWs training on packages, study participant age, the sample size, annual deworming programs, predisposing factors, and differences in the endemicity of parasites in the study areas.

Regarding species-specific report, this study showed a lower prevalence of *T. trichiura* (21.6%), *A. lumbricoides* (6.4%), and hookworm (2.3%) as compared to a studies conducted in Ecuadorian Amazon, Nigeria and Ethiopia with prevalence of 38% [24], 41.5 [27] and 31.3% [28] for *T. trichiura*, 48% [24], 19.5% [25], 37.2% [27], 67.7% [28] and 42% [30] for *A. lumbricoides*, and 7.6% [25], 28.4% [27], 45% [28] and 47% [30] for hookworm, respectively. However, a relatively lower prevalence of *T. trichiura* was observed elsewhere with 18.9% [25] and 11% [30]. Even though its species composition and occurrence vary, poly-parasitism is also common in many tropical and sub-tropical regions of the world in which the maximum

**Table 3. Bivariate and multivariate logistic regression of risk factors of STHs infection among individuals in model and non-model HHs, Seka Chekorsa woreda, Jimma zone, Southwest Ethiopia.**

| Variables | STHs +ve (n = 198) No (%) | COR (95% CI) | p-value | AOR (95% CI) | p-value |
|---|---|---|---|---|---|
| **HH status** | | | | | |
| Model | 45 (22.7) | Ref | | Ref | |
| Non-model | 153 (77.3) | 5.80 (3.94–8.55) | <0.001* | 6.54 (4.36–9.82) | <0.001[a] |
| **Gender** | | | | | |
| **Male** | 96 (48.5) | 0.89 (0.64–1.26) | 0.529 | | |
| Female | 102 (51.5) | Ref | | | |
| **Age** | | | | | |
| < 15 | 97 (49.0) | 1.39 (0.99–1.96) | 0.057* | 1.52 (1.04–2.21) | 0.030[b] |
| > 15 | 101 (51.0) | Ref | | Ref | |
| **Open defecation free (ODF)** | | | | | |
| Yes | 177 (89.4) | Ref | | Ref | |
| No | 21 (10.6) | 1.77 (0.97–3.23) | 0.060* | 1.23 (0.23–6.53) | 0.807 |
| **Latrine availability** | | | | | |
| Yes | 190 (96.0) | Ref | | Ref | |
| No | 8 (4.0) | 3.44 (1.11–10.67) | 0.032* | 3.33 (0.93–11.91) | 0.065 |
| **Latrine lid covered** | | | | | |
| Yes | 116 (58.6) | Ref | | Ref | |
| No | 82 (41.4) | 1.75 (1.23–2.49) | 0.002* | 1.07 (0.56–2.04) | 0.843 |
| **Water source** | | | | | |
| Pipe | 14 (7.1) | Ref | | | |
| Spring | 184 (92.9) | 1.41 (0.75–2.65) | 0.292 | | |
| **Hand washing facility** | | | | | |
| Yes | 137 (69.2) | Ref | | Ref | |
| No | 61 (30.8) | 1.42 (0.97–2.07) | 0.070* | 0.76 (0.48–1.18) | 0.221 |
| **Hand washing habit** | | | | | |
| Yes | 182 (91.9) | Ref | | Ref | |
| No | 16 (8.1) | 1.83 (0.92–3.64) | 0.086* | 0.69 (0.27–1.80) | 0.448 |
| **Shoe wearing** | | | | | |
| Yes | 178 (89.9) | Ref | | Ref | |
| No | 20 (10.1) | 1.75 (0.95–3.23) | 0.075* | 1.89 (0.65–5.53) | 0.243 |
| **Nail hygiene** | | | | | |
| No | 182 (91.9) | 3.22 (1.47–7.08) | 0.004* | 1.89 (0.77–4.62) | 0.164 |
| Clean | 16 (8.1) | Ref | | Ref | |
| **Waste disposal** | | | | | |
| Open field | 115 (58.1) | 2.05 (1.45–2.89) | <0.001* | 1.23 (0.833–1.82) | 0.298 |
| Pit | 83 (41.9) | Ref | | Ref | |
| **Latrine use pattern** | | | | | |
| Always | 178 (89.9) | Ref | | Ref | |
| Sometimes | 20 (10.1) | 1.75 (0.95–3.23) | 0.075* | 2.85 (1.43–5.69) | 0.003[b] |

Key: Ref: Reference category

*: Potential candidates for multivariate analysis p<0.25

[a]: P<0.001

[b]: p<0.05 considered as significantly associated with the dependent variable.

number of parasite species found in the same host in our study was three, and the prevalence of poly-parasite infection was 2.29%. This result is lower when compared with the studies reporting 8.66% [31], 56.7% [32] and 12.4% [20]. Variations in distribution and occurrence of STHs infections in different localities might be due to environmental, socio-demographic, and socio-economic factors that favor the transmission cycle of the parasites, sample size, egg output variation, and diagnostic technique performed.

The intensity of STHs infection ranges from light to moderate for individuals in model and non-model HHs except light infection intensity for hookworm species in both HHs. The intensity of STHs infection of individuals between the two HHs showed a significant difference for *A. lumbricoides* and *T. trichiura* while no significant difference for Hookworm species. This may be associated with the frequent exposure status of individuals among non-model HHs to the source of infection becoming higher. It could be related with the significant difference in their income status, latrine had coverlid and hand washing facility, and waste disposal mechanisms reducing risk factors. In the study done in the Ecuadorian Amazon, most of the individuals were infected with *A. lumbricoides* with a moderate intensity (51%) which was higher but 4.0% had a heavy intensity of infections. Similarly, most of the individuals infected with *T. trichiura* had light intensity (91.0%) [24] which is a lower intensity infection than the present study (96.5%). The intensity level difference may be due to frequent exposure status to the source of infection, the difference in treatment-seeking behavior, the difference in awareness of study participants, risk groups with age, and immunological differences.

Moreover, the current study has identified potential risk factors for STHs infection. Accordingly, age, latrine use pattern, and HH status (model or non-model) were found to be significant predictors of STHs infections in our study. Study participants, who are lower than 15 years were 1.5 times more likely to be infected by STHs than those who were greater than 15 years old. This is probably related to lower age groups having more exposure to soil and unhygienic practices. Similarly, individuals who use latrines sometimes were 2.85 more infected than individuals who use latrines always. This may be related to the possibility of open field defecation practice whenever the latrine is not used increasing the contamination with the parasite's infective stages from the soil. The HH status in our study was found to be the other potential risk factor for STHs infection as individuals in the non-model HHs were 6.5 times more infected than the model HHs. This may be related to the impact of the training package on improving the lifestyle of hygienic practice, awareness of infectious diseases transmission mode, health care service-seeking practices or the existing economic status differences among the HHs.

Among the HEWs training package trained HHs, the majority of the HHs (65%) knew about STHs and heard the information from health extension workers (50%) and health institutions (4.2%). This was slightly higher as compared to participants of Orang Asli in rural Malaysia, (61.4%) of HH's main source of information was the clinic while the majority of them could not remember the source [13]. The difference with the present study may be due to the level of awareness of health information increased with the efforts made by the health extension workers implementation of training packages.

## Conclusion

The prevalence of STHs infection reported among individual members in the model HHs showed a significant reduction and a significant difference in the intensity of infection when compared to non-model HHs. The HH status, age of study subjects, and latrine use pattern were found significant predictors of STHs infection prevalence in the study area. Poor personal-related hygiene and environmental contamination are still public health problems

among the individual members in the non-model HHs which can increase frequent exposure. Therefore, the application of health extension programs in communities should be strengthened sustainably in engaging all family members during the provision of HSEPs. Awareness creation on personal hygiene, lifestyle, and environmental sanitation-related infections such as STHs has to be strengthened along with deworming programs considering the adult community members. Such program implementation studies would better be assessed using a cluster randomized controlled trial study design to rule out the impact of a program to justify the outcome in controlled situations.

## Limitations

In the absence of baseline data for STH infection status during program implementation by the ministry of health of Ethiopia, we performed an observational study design after fifteen years of program launch to assess the difference in prevalence and intensity of STHs infection among individuals residing within the two groups of households. Besides, our selected households were also certified by the HEWs for fulfilling minimum requirements of the training packages and we cannot guarantee their sustainable implementation of training packages after their certification. We expect the enrolment and certification of the households to be based on ground level measurable indicator parameters regardless of the economic status or lifestyle improvement of the households. Furthermore, our finding was based on a small sample size and having a study design limitation for assessing the risk of exposure to acquiring STHs infection that might affect our results.

## Acknowledgments

We are thankful for the support and cooperation of the Jimma Zonal and Seka Woreda health offices, District administrators, health extension workers, and the participation of the selected communities and individuals within selected households.

## Author Contributions

**Conceptualization:** Yonas Alemu, Teshome Degefa, Mitiku Bajiro.

**Data curation:** Getachew Teshome.

**Formal analysis:** Getachew Teshome.

**Investigation:** Teshome Degefa, Getachew Teshome.

**Methodology:** Yonas Alemu, Teshome Degefa, Mitiku Bajiro.

**Project administration:** Yonas Alemu.

**Resources:** Getachew Teshome.

**Software:** Getachew Teshome.

**Supervision:** Yonas Alemu, Teshome Degefa, Mitiku Bajiro.

**Validation:** Yonas Alemu, Teshome Degefa, Mitiku Bajiro.

**Visualization:** Yonas Alemu, Teshome Degefa, Mitiku Bajiro.

**Writing – original draft:** Yonas Alemu, Getachew Teshome.

**Writing – review & editing:** Yonas Alemu, Teshome Degefa, Mitiku Bajiro, Getachew Teshome.

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
