## [Decision Letter · Decision Letter 0]

24 Feb 2022

PONE-D-21-35319Prevalence and Intensity of Soil-Transmitted Helminthes Infection among Model and Non-Model Households, South West Ethiopia: A Comparative Cross Sectional Community Based StudyPLOS ONE

Dear Dr. Alemu,

Thank you for submitting your manuscript to PLOS ONE. After careful consideration, we feel that it has merit but does not fully meet PLOS ONE’s publication criteria as it currently stands. Therefore, we invite you to submit a revised version of the manuscript that addresses the points raised during the review process.

We look forward to receiving your revised manuscript.

Kind regards,

Clement Ameh Yaro, Ph.D

Academic Editor

PLOS ONE

Journal Requirements:

Additional Editor Comments (if provided):

EDITOR’S COMMENTS

Dear Authors, kindly respond adequately to the points raised by the reviewers. There is need for improvement in the language structure. Clear explanations should be given on the study design and selection of study participants.

COMMENTS TO AUTHORS

FIRST REVIEWER:

First of all, I would appreciate and thanks authors for their interest and work to address evidence gaps on neglected tropical diseases.

General comments:

--I would expect specific target groups in the title and I suggest the title to be shaped as “Prevalence and Intensity of Soil-Transmitted Helminths Infection among individuals in Model and Non-Model Households, South West Ethiopia: A Comparative Cross Sectional Community Based Study”

-There is inconsistent use of the word STH and STHs in the manuscript.

- Given the main focus of your study is on STHs, I was wondering about the relevance of including other intestinal parasites in this study since

-How did authors categorize households as model and non-model?

-Lines 153-154, how was the wet mount examination performed (how specimens collected, transported, and where performed?).

-The sampling technique need to be clearly stated. Explanation for inclusion criteria of study participants should be stated y in the study?

-I suggest the word education (literate/illiterate) in your manuscript to be categorized by using other references? From my point of view, I don’t recommend using the word ‘illiterate’.

-The age category is not consistent in the manuscript; please revisit in the results section.

- The entire manuscript should be checked and proofread; there are language uses, typographical errors, and grammatical lapses that can be very distracting.

-To strengthen the paper more, the authors can suggest concrete recommendations on how the gaps in the program can be addressed.

-Authors suggestion concerning others operational research would be very important.

Specific comments:

The abstract should:

-Before stating the evidence gaps exist on the effect of modeling households on prevalence of soil-transmitted helminths among individuals, abstract should high light brief information about soil-transmitted helminths for readers outside the filed?

-Authors should mention the place where the study was done.

Methods:

- It seems that the authors aimed at testing the hypothesis of ‘individuals in non-model would have a higher prevalence of sol-transmitted helminths than individuals in the model households’. And by taking this into consideration, I was wondering how a cross-sectional study can testify this concept? For such kinds of study, I thought cohort study design would a good option to assess the effect of exposure on the outcome.

-How study participants were selected? Further explanation should be stated on how participants are selected to get a representative study population from model and non-model households? Did author consider stratification in the sampling technique.

--The inclusion criteria should be stated in the study population section.

-Did you include children under 15 years old in the study? State if assent was obtained from children in the ethical statement.

-The laboratory technique should be mentioned and explained as part of the data collection procedure.

- Please describe all the variables that you collected data on in the methods section.

-The data analysis procedure was not well stated and presented. The multivariate logistic regression model considered model and non-model households as one variable; by taking this into consideration how would author made a comparative analysis between those groups to draw a conclusion based on the data presented in the study.

-Please mention the ethical approval number if the study was reviewed and approved.

Results, discussion, and conclusions:

-Explain how the results related to the hypothesis presented as the basis of the study.

- The discussion provides a concise explanation of the implications of the findings, particularly in relation to previous related studies and potential future directions for research. Results should not be restated in the discussion section.

-The conclusion should be drawn based on the data presented in the study.

References:

-Please check all references according to the journals’ guidelines.

SECOND REVIEWER:

The title is very interesting and the manuscript technically sound, and the data support the conclusions. The analysis sounds to have significance on the prevention and control of STHs infection. The data underlying the findings in the manuscript fully available. The language that the authors used to write this manuscript is good. So with minor correction it can be considered for publication

Reviewers' comments:

Reviewer's Responses to Questions

**Comments to the Author**

1. Is the manuscript technically sound, and do the data support the conclusions?

Reviewer #1: Yes

Reviewer #2: Partly

2. Has the statistical analysis been performed appropriately and rigorously? 

Reviewer #1: Yes

Reviewer #2: No

3. Have the authors made all data underlying the findings in their manuscript fully available?

Reviewer #1: Yes

Reviewer #2: Yes

4. Is the manuscript presented in an intelligible fashion and written in standard English?

Reviewer #1: Yes

Reviewer #2: No

5. Review Comments to the Author

Reviewer #1: the title is very interesting and the manuscript technically sound, and the data support the conclusions. the analysis sounds to have significance on the prevention and control of STHs infection. the data underlying the findings in the manuscript fully available. the language that the authors used to write this manuscript is good. so with minor correction it can be considered for publication

Reviewer #2: First of all, I would appreciate and thanks authors for their interest and work to address evidence gaps on neglected tropical diseases.

General comments:

--I would expect specific target groups in the title and I suggest the title to be shaped as “Prevalence and Intensity of Soil-Transmitted Helminths Infection among individuals in Model and Non-Model Households, South West Ethiopia: A Comparative Cross Sectional Community Based Study”

-There is inconsistent use of the word STH and STHs in the manuscript.

- Given the main focus of your study is on STHs, I was wondering about the relevance of including other intestinal parasites in this study since

-How did authors categorize households as model and non-model?

-Lines 153-154, how was the wet mount examination performed (how specimens collected, transported, and where performed?).

-The sampling technique need to be clearly stated. Explanation for inclusion criteria of study participants should be stated y in the study?

-I suggest the word education (literate/illiterate) in your manuscript to be categorized by using other references? From my point of view, I don’t recommend using the word ‘illiterate’.

-The age category is not consistent in the manuscript; please revisit in the results section.

- The entire manuscript should be checked and proofread; there are language uses, typographical errors, and grammatical lapses that can be very distracting.

-To strengthen the paper more, the authors can suggest concrete recommendations on how the gaps in the program can be addressed.

-Authors suggestion concerning others operational research would be very important.

Specific comments:

The abstract should:

-Before stating the evidence gaps exist on the effect of modeling households on prevalence of soil-transmitted helminths among individuals, abstract should high light brief information about soil-transmitted helminths for readers outside the filed?

-Authors should mention the place where the study was done.

Methods:

- It seems that the authors aimed at testing the hypothesis of ‘individuals in non-model would have a higher prevalence of sol-transmitted helminths than individuals in the model households’. And by taking this into consideration, I was wondering how a cross-sectional study can testify this concept? For such kinds of study, I thought cohort study design would a good option to assess the effect of exposure on the outcome.

-How study participants were selected? Further explanation should be stated on how participants are selected to get a representative study population from model and non-model households? Did author consider stratification in the sampling technique.

--The inclusion criteria should be stated in the study population section.

-Did you include children under 15 years old in the study? State if assent was obtained from children in the ethical statement.

-The laboratory technique should be mentioned and explained as part of the data collection procedure.

- Please describe all the variables that you collected data on in the methods section.

-The data analysis procedure was not well stated and presented. The multivariate logistic regression model considered model and non-model households as one variable; by taking this into consideration how would author made a comparative analysis between those groups to draw a conclusion based on the data presented in the study.

-Please mention the ethical approval number if the study was reviewed and approved.

Results, discussion, and conclusions:

-Explain how the results related to the hypothesis presented as the basis of the study.

- The discussion provides a concise explanation of the implications of the findings, particularly in relation to previous related studies and potential future directions for research. Results should not be restated in the discussion section.

-The conclusion should be drawn based on the data presented in the study.

References:

-Please check all references according to the journals’ guidelines.

6. PLOS authors have the option to publish the peer review history of their article (what does this mean?). If published, this will include your full peer review and any attached files.

Reviewer #1: No

Reviewer #2: No

---

## [Author Response · Author response to Decision Letter 0]

15 Apr 2022

EDITOR’S COMMENTS

Dear Authors, kindly respond adequately to the points raised by the reviewers. There is need for improvement in the language structure. Clear explanations should be given on the selection of study participants, and study design used.

(First of all, we would like to thank the editor handling our manuscript. We also extend our appreciation to the reviewers who provided concrete comments that can help our write-up to progress one step forward. We have accepted and revised the manuscript incorporating the comments given by the reviewers. Our responses to some of the questions are also incorporated in the comment reviewing pane).

COMMENTS TO AUTHORS

FIRST REVIEWER:

First of all, I would appreciate and thanks authors for their interest and work to address evidence gaps on neglected tropical diseases.

(We would like to thank the first reviewer, for the positive and constructive comment appreciating our focus area which tried to show evidence of STH prevalence and intensity difference among communities targeted with a health extension training program package and its impact on one of the NTDs, which is STHs).

General comments:

--I would expect specific target groups in the title and I suggest the title to be shaped as “Prevalence and Intensity of Soil-Transmitted Helminths Infection among individuals in Model and Non-Model Households, South West Ethiopia: A Comparative Cross Sectional Community Based Study”

(We would like to thank the first reviewer, for the positive and constructive comment appreciating our focus area which tried to show evidence of STH prevalence and intensity difference among communities targeted with a health extension training program package and its impact on one of the NTDs, which is STHs).

-There is inconsistent use of the word STH and STHs in the manuscript.

(Thank you, we made a correction).

- Given the main focus of your study is on STHs, I was wondering about the relevance of including other intestinal parasites in this study since

(Sure, our main focus is to show the prevalence of STH among our study subjects. Since we used saline wet mount and Kato Katz techniques, other protozoans and helminths parasites were detected in the stool of the study participants along with the STHs. 

However, we have no special interest to display other IP prevalence except to show there were multiple parasite infections as an additional finding. We will remove it from the document if suggested not relevant).

-How did authors categorize households as model and non-model?

(Thanks, we did not categorize the households into model and non-model. This category was already made by the program criteria as model household heads (members) who attended at least 75% of training packages of health service extension programs (HSEPs), implemented at least 75% of the packages, and were eventually certified for fulfilling these requirements by the government. Otherwise, they were labeled as non-model in the program. Thus, we were aware of the status confirmed from their certificates).

-Lines 153-154, how was the wet mount examination performed (how specimens collected, transported, and where performed?).

(Thanks, Each selected Kebele has at least one health post or an additional health center according to the population in the woreda/district, and also we were traveling having microscopy for the saline wet mount exam. This makes it nearby to the households to be examined within 30 minutes. The collected stool specimen in a stool cup was then transported to Jimma university research laboratory in a cold box within 2 hours of collection for the Kato Katz procedure).

-The sampling technique need to be clearly stated. Explanation for inclusion criteria of study participants should be stated y in the study?

(Thank you for the comment, we rewrote the sampling technique including diagrammatic representation, and added eligibility criteria in the study population section).

-I suggest the word education (literate/illiterate) in your manuscript to be categorized by using other references? From my point of view, I don’t recommend using the word ‘illiterate’.

(Thank you, we have changed to read and write for literate and unable to read and write for the illiterate).

-The age category is not consistent in the manuscript; please revisit in the results section.

(Thank you for the comment, we have reviewed the result section. Our study subjects are the HH heads and members of the HH which makes a range from children to adults. We consider the household level question for table 1, based on their exposure to STH intensity in table 2 and risked age group in table 3. However, now we made a correction to remove ambiguity and explained relevant findings in text only).

- The entire manuscript should be checked and proofread; there are language uses, typographical errors, and grammatical lapses that can be very distracting.

(We appreciate the comment, we have rechecked the whole document for the language, typographic, and Grammatik issues and made corrections in the main text).

-To strengthen the paper more, the authors can suggest concrete recommendations on how the gaps in the program can be addressed.

-Authors suggestion concerning others operational research would be very important.

(Thank you, we have included our recommendation with a suggestion for filling gaps and strengthening the program sustainably in the community).

Specific comments:

The abstract should:

-Before stating the evidence gaps exist on the effect of modeling households on prevalence of soil-transmitted helminths among individuals, abstract should high light brief information about soil-transmitted helminths for readers outside the filed?

(Thank you, we have incorporated introductory information on STH in the abstract section).

-Authors should mention the place where the study was done.

(Thank you, we have incorporated the study area in the abstract).

Methods:

- It seems that the authors aimed at testing the hypothesis of ‘individuals in non-model would have a higher prevalence of sol-transmitted helminths than individuals in the model households’. And by taking this into consideration, I was wondering how a cross-sectional study can testify this concept? For such kinds of study, I thought cohort study design would a good option to assess the effect of exposure on the outcome.

(Thank you for the comment, we understand the suggested cohort design could be more powerful to justify the situation. Also, the baseline assessment was not done while the program was implemented in 2003 to at least look at the trend recommending at least baseline assessments to be performed during launching.

We aimed to assess the impact of the program just by looking at the difference in the prevalence, intensity, and risk factors associated among the individuals from model and non-model HHs at the point of time and provide evidence for further research questions as baseline information).

-How study participants were selected? Further explanation should be stated on how participants are selected to get a representative study population from model and non-model households? Did author consider stratification in the sampling technique.

(Thank you, we have incorporated the comment and explained in the main text. We made a selection starting from the Kebele/districts from the purposely selected Seka chekorsa woreda in the Jimma zone. Then villages/zones were included w/c is located in the district. From the villages, HHs were systematically selected. Finally, all eligible individuals within the HHs were surveyed in our study. Terms like woreda, district, and villages are sequential local administrative strata as a subset of the other respectively).

--The inclusion criteria should be stated in the study population section.

(Yes, we have incorporated it in the mentioned section).

-Did you include children under 15 years old in the study? State if assent was obtained from children in the ethical statement.

(Sure, we have included children aged less than 15 years and mentioned now as their assent was taken).

-The laboratory technique should be mentioned and explained as part of the data collection procedure.

(Thank you, we have explained the technique employed for the stool examination. If the detailed procedure of wet mount examination and Kato Katz techniques are requested, we may add additional documents to the supplementary file).

- Please describe all the variables that you collected data on in the methods section.

(Thank you, we have listed the summary of variables included in our study in the methods section).

-The data analysis procedure was not well stated and presented. The multivariate logistic regression model considered model and non-model households as one variable; by taking this into consideration how would author made a comparative analysis between those groups to draw a conclusion based on the data presented in the study.

(Thank you, we made this clear under the data analysis now. Multivariable analysis was made to see the association of household status (either model or non-model) to STHs infection. This is to give evidence of whether the non-model households have more risk of being infected with STHs. But comparisons were made between the two groups using an independent sample t-test).

-Please mention the ethical approval number if the study was reviewed and approved.

(Thank you, we have mentioned the ethical approval number given by the institutional research ethical review board).

Results, discussion, and conclusions:

-Explain how the results related to the hypothesis presented as the basis of the study.

(Thank you, our results aimed to show the individuals in the model or non-model HHs were compared for the difference in prevalence, and intensity of STHs infection. We re-wrote our result section based on the comments which may clear ambiguity now).

- The discussion provides a concise explanation of the implications of the findings, particularly in relation to previous related studies and potential future directions for research. Results should not be restated in the discussion section.

(Thank you, our results aimed to show the individuals in the model or non-model HHs were compared for the difference in prevalence, and intensity of STHs infection. We re-wrote our result section based on the comments which may clear ambiguity now).

-The conclusion should be drawn based on the data presented in the study.

(Thank you, we re-wrote our conclusion based on our result findings).

References:

-Please check all references according to the journals’ guidelines.

(Sure, references were scanned again to fit the guidelines of the journal).

SECOND REVIEWER:

The title is very interesting and the manuscript technically sound, and the data support the conclusions. The analysis sounds to have significance on the prevention and control of STHs infection. The data underlying the findings in the manuscript fully available. The language that the authors used to write this manuscript is good. So with minor correction it can be considered for publication

(Thank you for the second reviewer, we would like to extend our appreciation for the positive and constructive comment. We have updated the write-up based on the provided comments and hope it will be considered for further process of publication).

---

## [Decision Letter · Decision Letter 1]

17 Jun 2022

PONE-D-21-35319R1

Prevalence and Intensity of Soil-Transmitted Helminths Infection among individuals in Model and Non-Model Households, South West Ethiopia: A Comparative Cross-Sectional Community Based Study

PLOS ONE

Dear Dr. Alemu,

Thank you for submitting your manuscript to PLOS ONE. After careful consideration, we feel that it has merit but does not fully meet PLOS ONE’s publication criteria as it currently stands. Therefore, we invite you to submit a revised version of the manuscript that addresses the points raised during the review process.

We look forward to receiving your revised manuscript.

Kind regards,

Clement Ameh Yaro, Ph.D

Academic Editor

PLOS ONE

Journal Requirements:

Additional Editor Comments (if provided):

Third Reviewer:

I was not the reviewer of the first version of the manuscript, for this reason I raised here issues that were not mentioned during the first revision (I am sorry for that)

I think the manuscript is interesting and merit publication but need some important revision including recalculation of the geometric means:

1 in the Material and method the authors need to clearly specify how the geometric mean was calculated.

From line 266 It is my impression that the mean was calculated only for the positive samples; for example for T trichiura (since several negative individuals were identified) the range of the egg count should be between 0- and 11 104 and not between 48 and 11 104.

To compare the two communities (in this case model and not-model) is important to calculate the mean epg in the entire community and not only on the positive cases (and to do so is necessary to include the zero counts in the calculation of the mean epg) .

calculating mean epg is only the positive cases could result in poor interpretation of the data:

Please consider the following hypothetical and extreme:

- Only a single individual is identified as positive in the model group and this patient has high epg

- All the individuals are positive in the non-model group with different level of epg

If the mean epg is calculated on positive only (as it is done now by the author) the result is that in the model group the intensity of infection is much higher than in the non -model

If the epg is calculated on the entire model group (including negative patients), because the high number of zero the epg will be much lower than in the non-model and the authors will correctly conclude that the intensity of infection is lower in the model group.

I realize the difficulties in calculating the geometric mean including 0 (the logarithm of 0 is not defined) but there are mathematical way to do this (adding 1 to each egg count end then removing 1 from the geometric mean) or more simply by using arithmetic mean.

I think the statement in line 273-274 about the mean intensity of STH infection in model ad non model should be revised if the mean is calculated including the zero counts.

2 the authors in the discussion should mention some important limitation of the study

a. the very small sample size

b. the fact that the family income was much higher in the model than in the non-model community (this is also reflected in better latrine and hand washing facilities (see table1). The better financial situation of the model families could have been the reason (confounding factor) of the lower prevalence and intensity of STH infections and not the activities conducted in the model group.

Reviewers' comments:

Reviewer's Responses to Questions

**Comments to the Author**

1. If the authors have adequately addressed your comments raised in a previous round of review and you feel that this manuscript is now acceptable for publication, you may indicate that here to bypass the “Comments to the Author” section, enter your conflict of interest statement in the “Confidential to Editor” section, and submit your "Accept" recommendation.

Reviewer #2: All comments have been addressed

Reviewer #3: All comments have been addressed

Reviewer #4: (No Response)

2. Is the manuscript technically sound, and do the data support the conclusions?

Reviewer #2: Yes

Reviewer #3: (No Response)

Reviewer #4: No

3. Has the statistical analysis been performed appropriately and rigorously? 

Reviewer #2: Yes

Reviewer #3: (No Response)

Reviewer #4: I Don't Know

4. Have the authors made all data underlying the findings in their manuscript fully available?

Reviewer #2: Yes

Reviewer #3: (No Response)

Reviewer #4: No

5. Is the manuscript presented in an intelligible fashion and written in standard English?

Reviewer #2: Yes

Reviewer #3: (No Response)

Reviewer #4: No

6. Review Comments to the Author

Reviewer #2: Authors made substantial revision to address the comments.

Few comments:

Please check journal publication style, format, and referencing (for instance, check line 317).

Concerning the discussion, explanation and recommendations on implication of the findings could improve the manuscript, instead of giving more emphasis on comparing the study findings with findings of other studies.

Thank you.

Reviewer #3: I was not the reviewer of the first version of the manuscript, for this reason I raised here issues that were not mentioned during the first revision (I am sorry for that)

I think the manuscript is interesting and merit publication but need some important revision including recalculation of the geometric means:

1 in the Material and method the authors need to clearly specify how the geometric mean was calculated.

From line 266 It is my impression that the mean was calculated only for the positive samples; for example for T trichiura (since several negative individuals were identified) the range of the egg count should be between 0- and 11 104 and not between 48 and 11 104.

To compare the two communities (in this case model and not-model) is important to calculate the mean epg in the entire community and not only on the positive cases (and to do so is necessary to include the zero counts in the calculation of the mean epg) .

calculating mean epg is only the positive cases could result in poor interpretation of the data:

Please consider the following hypothetical and extreme:

- Only a single individual is identified as positive in the model group and this patient has high epg

- All the individuals are positive in the non-model group with different level of epg

If the mean epg is calculated on positive only (as it is done now by the author) the result is that in the model group the intensity of infection is much higher than in the non -model

If the epg is calculated on the entire model group (including negative patients), because the high number of zero the epg will be much lower than in the non-model and the authors will correctly conclude that the intensity of infection is lower in the model group.

I realize the difficulties in calculating the geometric mean including 0 (the logarithm of 0 is not defined) but there are mathematical way to do this (adding 1 to each egg count end then removing 1 from the geometric mean) or more simply by using arithmetic mean.

I think the statement in line 273-274 about the mean intensity of STH infection in model ad non model should be revised if the mean is calculated including the zero counts.

2 the authors in the discussion should mention some important limitation of the study

a. the very small sample size

b. the fact that the family income was much higher in the model than in the non-model community (this is also reflected in better latrine and hand washing facilities (see table1). The better financial situation of the model families could have been the reason (confounding factor) of the lower prevalence and intensity of STH infections and not the activities conducted in the model group.

Reviewer #4: The manuscript addresses an important group of the neglected tropical diseases (NTDs), but has several shortcomings as stated below.

Grammar: The manuscript has grammatical mistakes which will require to be addressed. For example, the opening statement in the abstract should simply state that soil-transmitted helminths (STH) is a term used to refer to infections caused by intestinal worms which are transmitted through contaminated soil.

Study design: The investigation was an observational study with a major methodological shortcoming because baseline data were not collected. It is therefore very difficult to assess the impact of the public health interventions provided to the households particularly when the primary endpoints are prevalence and intensity of STH infections. Ideally, the study should have employed a cluster randomized controlled trial design.

Results: The results presented should be taken with caution because of the methodological challenge mentioned in the study design.

Discussion: Study limitations are not explicitly highlighted and their implications on the results discussed adequately.

7. PLOS authors have the option to publish the peer review history of their article (what does this mean?). If published, this will include your full peer review and any attached files.

Reviewer #2: No

Reviewer #3: No

Reviewer #4: No

---

## [Author Response · Author response to Decision Letter 1]

29 Jul 2022

Response to the Reviewers and editor: 

We would like to extend our gratitude to the editor and reviewers in charge of evaluating our manuscript for fitting the journals publishing requirements.

• We have checked the references list; however, we have no article retracted from the journals.

• Thank you the third reviewer for finding our manuscript interesting and could have merit to the field, we appreciate. 

• We accept the critical and important comment raised on the mean calculation. However, we prefer to use the arithmetic mean calculation for the sake of simplicity and we have made corrections accordingly such as including zero intensity counts in negative individuals to calculate the mean and updating in the text. 

• We made also similar corrections in the main text indicating the way data was analyzed for the methods section. 

• We have added the limitation of our study under the discussion section separately.

---

## [Decision Letter · Decision Letter 2]

7 Sep 2022

PONE-D-21-35319R2Prevalence and Intensity of Soil-Transmitted Helminths Infection among individuals in Model and Non-Model Households, South West Ethiopia: A Comparative Cross-Sectional Community Based StudyPLOS ONE

Dear Dr. Alemu,

Thank you for submitting your manuscript to PLOS ONE. After careful consideration, we feel that it has merit but does not fully meet PLOS ONE’s publication criteria as it currently stands. Therefore, we invite you to submit a revised version of the manuscript that addresses the points raised during the review process.

Kindly attend to these comments if you wish the manuscript to be considered for publication.

- the calculation of the geometric mean one and the request to explain if the zero counts were included in the mean.

- the second on the request to mention the limitation of the study.

Also, the second reviewer requested the following minor corrections;

1.     It is important to make it clear what the abbreviation HH is referring to because it appears to be interchangeably used to refer to, 1) household heads, and/or 2) households.

2.     Line 24-27: Check grammar for the statement reading “This study…(HEP)”

3.     Line 37: Insert the word “to be” between the words found and 32.4%.

We look forward to receiving your revised manuscript.

Kind regards,

Clement Ameh Yaro, Ph.D

Academic Editor 

PLOS ONE

Journal Requirements:

Additional Editor Comments (if provided):

Dear Authors, kindly attend to these comments if you wish the manuscript to be considered for publication.

- the calculation of the geometric mean one and the request to explain if the zero counts were included in the mean.

- the second on the request to mention the limitation of the study.

Also, the second reviewer requested the following minor corrections;

1. It is important to make it clear what the abbreviation HH is referring to because it appears to be interchangeably used to refer to, 1) household heads, and/or 2) households.

2. Line 24-27: Check grammar for the statement reading “This study…(HEP)”

3. Line 37: Insert the word “to be” between the words found and 32.4%.

Reviewers' comments:

Reviewer's Responses to Questions

**Comments to the Author**

1. If the authors have adequately addressed your comments raised in a previous round of review and you feel that this manuscript is now acceptable for publication, you may indicate that here to bypass the “Comments to the Author” section, enter your conflict of interest statement in the “Confidential to Editor” section, and submit your "Accept" recommendation.

Reviewer #3: (No Response)

Reviewer #4: All comments have been addressed

2. Is the manuscript technically sound, and do the data support the conclusions?

Reviewer #3: Partly

Reviewer #4: Yes

3. Has the statistical analysis been performed appropriately and rigorously? 

Reviewer #3: No

Reviewer #4: Yes

4. Have the authors made all data underlying the findings in their manuscript fully available?

Reviewer #3: Yes

Reviewer #4: Yes

5. Is the manuscript presented in an intelligible fashion and written in standard English?

Reviewer #3: Yes

Reviewer #4: Yes

6. Review Comments to the Author

Reviewer #3: The authors ignored the two comments made to the previous version:

- one on the calculation if the geometric mean one and the request to explain if the zero counts were included in the mean

- the second on the request to mention the limitation of the study.

for this reason i can not agree on the revision

Reviewer #4: (No Response)

7. PLOS authors have the option to publish the peer review history of their article (what does this mean?). If published, this will include your full peer review and any attached files.

Reviewer #3: No

Reviewer #4: No

---

## [Author Response · Author response to Decision Letter 2]

15 Sep 2022

PONE-D-21-35319R2: Reviewer’s comments 

Response: 

(We would like to extend our appreciation to the editor handling our manuscripts review process once again)

Summary

The author have revised the manuscript (discussion) and highlighted the lack of baseline data as a major limitation. Thus, the manuscript may be considered for publication. There are a few minor comments below.

Response: 

(We are grateful to the reviewers for their critical comments and corrective suggestions in strengthening our manuscript to be considered for publication. We have responded to the recent comments accordingly in this version. 

In the previous version, we have tried to respond forwarding our revised version on raised comments related with data analysis and limitation of our study. To remind, we have incorporated the limitation of our study after the conclusion section of the main text in our former communication. 

Whereas, it seems that the issue related with the mean calculation remains unsatisfactory. As a result, we have revised our current version incorporating such comments as well. For the calculation of geometric mean, we have included zero counts in the mean calculation after adding one to each datasets and subtracting one from the result of converted logarithmic calculations)

Specific comments

1. It is important to make it clear what the abbreviation HH is referring to because it appears to be interchangeably used to refer to, 1) household heads, and/or 2) households. 

Response: 

(Thank you for the comment, we have rewritten based on their concept as HH for household, HHs for household heads and also identified separately in the text whenever it refers to the individual members in the HHs accordingly)

2. Line 24-27: Check grammar for the statement reading “This study…(HEP)”

Response:

(Thank you, We made adjustment to the text in response to your suggested comment)

3. Line 37: Insert the word “to be” between the words found and 32.4%. 

Response:

(Noted and we have incorporated the comment in the text)

---

## [Decision Letter · Decision Letter 3]

29 Sep 2022

Prevalence and Intensity of Soil-Transmitted Helminths Infection among individuals in Model and Non-Model Households, South West Ethiopia: A Comparative Cross-Sectional Community Based Study

PONE-D-21-35319R3

Dear Dr. Alemu,

We’re pleased to inform you that your manuscript has been judged scientifically suitable for publication and will be formally accepted for publication once it meets all outstanding technical requirements.

Kind regards,

Clement Ameh Yaro, Ph.D

Academic Editor

PLOS ONE

Additional Editor Comments (optional):

Reviewers' comments:

Reviewer's Responses to Questions

**Comments to the Author**

1. If the authors have adequately addressed your comments raised in a previous round of review and you feel that this manuscript is now acceptable for publication, you may indicate that here to bypass the “Comments to the Author” section, enter your conflict of interest statement in the “Confidential to Editor” section, and submit your "Accept" recommendation.

Reviewer #3: All comments have been addressed

Reviewer #4: All comments have been addressed

Reviewer #5: All comments have been addressed

2. Is the manuscript technically sound, and do the data support the conclusions?

Reviewer #3: (No Response)

Reviewer #4: Yes

Reviewer #5: Yes

3. Has the statistical analysis been performed appropriately and rigorously? 

Reviewer #3: I Don't Know

Reviewer #4: Yes

Reviewer #5: Yes

4. Have the authors made all data underlying the findings in their manuscript fully available?

Reviewer #3: Yes

Reviewer #4: Yes

Reviewer #5: Yes

5. Is the manuscript presented in an intelligible fashion and written in standard English?

Reviewer #3: Yes

Reviewer #4: Yes

Reviewer #5: Yes

6. Review Comments to the Author

Reviewer #3: (No Response)

Reviewer #4: (No Response)

Reviewer #5: authors have addressed the comments to my level of satisfaction, I have no further comments but to recommend accept for publication

7. PLOS authors have the option to publish the peer review history of their article (what does this mean?). If published, this will include your full peer review and any attached files.

Reviewer #3: No

Reviewer #4: No

Reviewer #5: No

---

## [Editor Report · Acceptance letter]

6 Oct 2022

PONE-D-21-35319R3 

Prevalence and Intensity of Soil-Transmitted Helminths Infection among individuals in Model and Non-Model Households, South West Ethiopia: A Comparative Cross-Sectional Community Based Study. 

Dear Dr. Alemu:

I'm pleased to inform you that your manuscript has been deemed suitable for publication in PLOS ONE. Congratulations! Your manuscript is now with our production department. 

Kind regards, 

on behalf of

Dr. Clement Ameh Yaro 

Academic Editor

PLOS ONE